# Unifying Dynamic Tool Creation and Cross-Task Experience Sharing through Cognitive Memory Architecture

## Abstract

Large Language Model agents face fundamental challenges in adapting to novel tasks due to limitations in tool availability and experience reuse. Existing approaches either rely on predefined tools with limited coverage or build tools from scratch without leveraging past experiences, leading to inefficient exploration and suboptimal performance. We introduce SMITH (Shared Memory Integrated Tool Hub), a unified cognitive architecture that seamlessly integrates dynamic tool creation with cross-task experience sharing through hierarchical memory organization. SMITH organizes agent memory into procedural, semantic, and episodic components, enabling systematic capability expansion while preserving successful execution patterns. Our approach formalizes tool creation as iterative code generation within controlled sandbox environments and experience sharing through episodic memory retrieval with semantic similarity matching. We further propose a curriculum learning strategy based on agent-ensemble difficulty re-estimation. Extensive experiments on the GAIA benchmark demonstrate SMITH's effectiveness, achieving 81.8% Pass@1 accuracy and outperforming state-of-the-art baselines including Alita (75.2%) and Memento (70.9%). Our work establishes a foundation for building truly adaptive agents that continuously evolve their capabilities through principled integration of tool creation and experience accumulation.

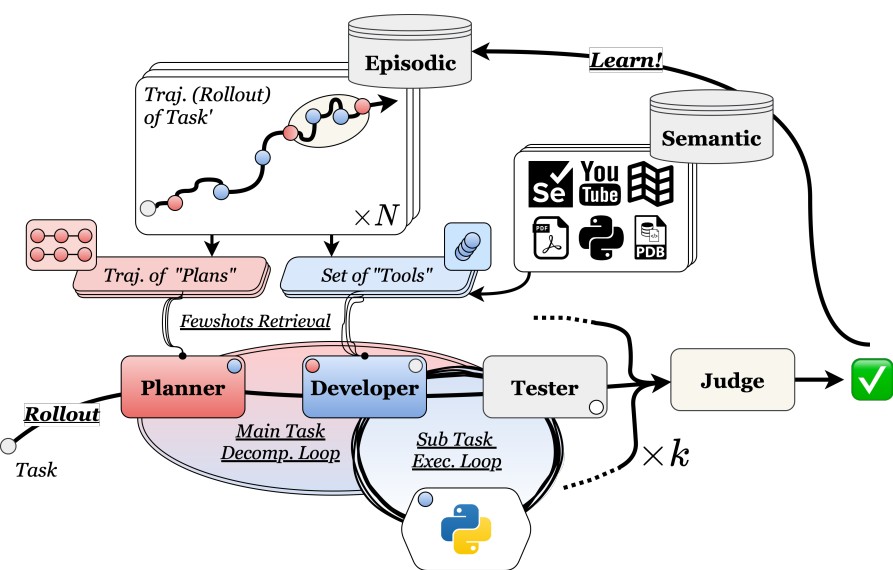

Figure 1: SMITH architecture overview. Each agent rollout involves two nested loops: an inner developer-tester loop for iterative code generation and debugging, and an outer planner loop for sub-plan execution. $\times k = 3$ represents 3-path sampling with LLM-as-a-judge consensus voting. Upon successful task completion, corresponding experiences are processed, embedded, and stored for future learning and reuse.

## 1 INTRODUCTION

The development of general AI assistants capable of tackling diverse, real-world tasks remains a fundamental challenge in artificial intelligence. While Large Language Models (LLM) have demonstrated remarkable reasoning capabilities, their application to complex problem-solving scenarios is often limited by two critical bottlenecks: the availability of appropriate tools for task execution and the ability to leverage past experiences for novel situations. Current approaches address these challenges in isolation—tool learning frameworks like Toolformer (Schick et al., 2023) rely on predefined tool collections with limited coverage, while recent tool creation methods such as Alita (Qiu et al., 2025) generate tools from scratch without systematic reuse. Similarly, experience sharing approaches like Memento (Zhou et al., 2025) focus on cross-task memory transfer but lack integrated tool creation capabilities. This fragmentation prevents agents from achieving the adaptive, cumulative learning characteristic of human problem-solving, where tools are created, refined, and reused across related tasks while successful strategies are systematically transferred to new domains.

We propose SMITH (Shared Memory Integrated Tool Hub), a unified cognitive architecture that bridges this gap by seamlessly integrating dynamic tool creation with cross-task experience sharing through a hierarchical memory framework. Drawing inspiration from cognitive architectures for language agents (Sumers et al., 2023), SMITH organizes agent memory into procedural, semantic, and episodic components, enabling systematic capability expansion while preserving successful execution patterns across tasks. Our approach formalizes tool creation as an iterative code generation process within controlled sandbox environments, and experience sharing through episodic memory retrieval with semantic similarity matching. To optimize learning efficiency, we introduce a novel curriculum learning strategy based on agent-ensemble difficulty re-estimation that reranks tasks according to agent-specific capability assessments rather than human annotations. This unified framework enables agents to continuously evolve their problem-solving capabilities through principled integration of tool creation and experience accumulation, establishing a foundation for truly adaptive AI systems that can tackle the complexity and diversity of real-world challenges.

## 2 RELATED WORK

**Multi-Agent Systems and General AI Assistants.** Benchmarks like GAIA (Mialon et al., 2023) evaluate general AI assistants through real-world questions requiring reasoning, multi-modality handling, and tool-use proficiency. AutoAgent (Tang et al., 2025) democratizes development through zero-code interfaces, OWL (Hu et al., 2025) enables cross-domain adaptation via hierarchical architectures, and AWorld (Yu et al., 2025) accelerates experience collection by 14.6× through distributed infrastructure. These approaches establish foundations for scalable, general-purpose AI assistants.

**Memory Architectures for Language Agents.** Context window limitations have driven extensive research into memory architectures for language agents. Building on memory networks (Weston et al., 2014) and retrieval-augmented generation (Lewis et al., 2020), Sumers et al. (2023) established theoretical foundations through Cognitive Architectures for LLM Agents, organizing agent memory into working, episodic, semantic, and procedural memory hierarchies. Practical implementations include MemGPT (Packer et al., 2023) with OS-inspired virtual context management, Mem0 (Chhikara et al., 2025) with graph-based representations, and self-controlled frameworks (Wang et al., 2023) achieving 77.1% accuracy with 91% lower latency. These advances enable persistent, context-aware systems like Generative Agents (Park et al., 2023) and ReAct (Yao et al., 2022).

**Tool Learning and Tool Creation.** While tool learning utilizes pre-existing tools (Schick et al., 2023), it requires human developers to design tools beforehand. Recent work shifted toward tool creation, enabling autonomous tool generation at runtime. Early methods like CRAFT (Yuan et al., 2024), CREATOR (Qian et al., 2023), and LATM (Cai et al., 2024) generated simple Python functions but lacked system interaction capabilities. Advanced frameworks expanded these capabilities: Wölflein et al. (2025) introduced ToolMaker for transforming scientific repositories into LLM-compatible tools, while Qiu et al. (2025) proposed Alita achieving 75.15% on GAIA through "minimal predefinition and maximal self-evolution" using Model Context Protocols. The key innovation, emerging from frameworks like SmolAgent (Roucher et al., 2025), is the ability to save and cache generated tools, forming closed-loop systems where successful tools become reusable assets.

**Experience Sharing.** Parameter-based methods like AWorld (Yu et al., 2025) and WebShaper (Tao et al., 2025) employ supervised fine-tuning followed by reinforcement learning but suffer from unclear memory hierarchies and inability to perform continual learning during inference. Memory-based approaches address these limitations by storing task execution memories as episodic traces without parameter modifications. Zhou et al. (2025) introduced Memento with Memory-augmented Markov Decision Process (M-MDP) achieving 87.88% Pass@3 on GAIA, Li et al. (2025) proposed MAEL for multi-agent cross-task experiential learning, and Yang et al. (2024) developed CoPS with pessimism-based experience selection. As noted in cognitive architectures (Sumers et al., 2023), experience sharing manages episodic memory through embedding-based retrieval systems with structural commonalities to memory management frameworks.

## 3 METHOD

### 3.1 FORMALIZATION OF DYNAMIC TOOL CREATION

We formalize the dynamic tool creation process as an interactive code generation and refinement procedure within a controlled execution environment. This formalization captures the iterative nature of tool development, where agents continuously write, test, debug, and refine code until successful tool implementation is achieved.

**Sandbox Environment and Agent Interaction.** We define a python sandbox execution environment $\langle \mathcal{E}, \texttt{exec}, \texttt{feedback} \rangle$ where $\mathcal{E}$ represents the current environment state, $\texttt{exec} : \mathcal{E} \times C \to \mathcal{E} \times O$ executes code $C$ and returns updated state and output $O$, and $\texttt{feedback} : O \to F$ provides structured error or success feedback $F$. Let agent $a$ represent the code-writing entity that interacts with environment through an iterative debugging loop. At each iteration $t$, the agent maintains code $c_t$ and receives feedback $f_t$ from the sandbox environment.

**Interactive Tool Creation Process.** Given a task specification $\tau$, the tool creation process unfolds as an iterative refinement sequence

$$c_{t+1} = \texttt{agent}(c_t, f_t, \tau, \mathcal{C}_{\text{code}}) \tag{1}$$

where $\mathcal{C}_{\text{code}} = \{(c_{t-1}, f_{t-1}), (c_{t-2}, f_{t-2}), \ldots\}$ represents contextual memory containing historical code-feedback pairs, debugging patterns, and successful implementation trajectories from previous iterations. The process continues until the sandbox environment returns successful feedback

$$f_t = \texttt{feedback}(\texttt{exec}(\mathcal{E}_t, c_t)) \in \{\checkmark, e_t\} \tag{2}$$

When $f_t = e_t$, the agent analyzes the error $e_t$ and generates refined code $c_{t+1}$. This debug-and-refine cycle continues until $f_t = \checkmark$. Upon successful execution, typically the code $c_{\text{done}}$ undergoes encapsulation to form a *tool*, or from a more comprehensive perspective, it can be formalized as a tool creation memory episode where the concept of *tool* is dissolved into past action execution

$$T = \{c_{done}, (c_0, \mathcal{E}_0, f_0) \ldots (c_{\text{done}}, \mathcal{E}_{\text{done}}, \checkmark)\} \tag{3}$$

where $(c_0, \mathcal{E}_0, f_0) \to (c_{\text{done}}, \mathcal{E}_{\text{done}}, \checkmark)$ captures the complete debugging trajectory. The tool repository $\mathbb{T}$ evolves dynamically as $\mathbb{T} \leftarrow \mathbb{T} \cup \{T\}$, enabling future tool reuse and composition.

### 3.2 FORMALIZATION OF CROSS-TASK EXPERIENCE SHARING

We formalize cross-task experience sharing through an episodic memory framework that enables agents to leverage previous successful execution patterns with semantically similarity. We establish the following assumption with $\mathcal{T} = \{\tau_1, \tau_2, \ldots, \tau_n\}$ denoting the task universe.

**Assumption 1 (Semantic Task Similarity)** *Two tasks $\tau_i, \tau_j \in \mathcal{T}$ are considered semantically similar if their problem structures and solution requirements exhibit similar patterns in semantic space, as measured by the similarity of their embedding representations $\Phi(\tau_i)$ and $\Phi(\tau_j)$. Formally, we define semantic similarity as $sim(\Phi(\tau_i), \Phi(\tau_j)) > \theta$ for some threshold $\theta$. Tasks satisfying this similarity criterion enable transferability of execution experiences across these tasks.*

Each agent $j$ maintains its own episodic memory $\mathcal{M}_{\text{ep}}^{(j)} = \{e_1^{(j)}, e_2^{(j)}, \ldots, e_k^{(j)}\}$, where each experience encapsulates a complete trajectory

$$e_i^{(j)} = \{\tau_l, (s_0, a_0) \ldots (s_{\text{done}}, a_{\text{done}})\} \tag{4}$$

where $s_t$ represents the agent's observation state at step $t$ (including task context, current progress, and environmental feedback), and $a_t$ denotes the action taken (e.g., code generation, tool invocation, or sub-plan decomposition).

We then define abstraction function $\Phi : e_i^{(j)} \to \mathbf{m}_i$ that varies by action space, where code writing actions require summarization before embedding, while planning agents perform intention decomposition and augmentation on proposed plans (details in Sec. 4).

**Experience Retrieval and Policy Enhancement.** Given current task $\tau$ and state $s_t$, agent node $j$ retrieves top-$k$ experiences via similarity scoring

$$r(e_i^{(j)}, \tau, s_t) = \langle \Phi(\{\tau, (s_t, \cdot)\}), \mathbf{m}_i \rangle \tag{5}$$

The top-$k$ experiences are retrieved as

$$m_t = \text{TopK}_{e_i^{(j)} \in \mathcal{M}_{\text{ep}}^{(j)}} r(e_i^{(j)}, \tau, s_t) \tag{6}$$

and actions are sampled as $a_t \sim \pi(\tau \oplus s_t \oplus m_t)$.

**Memory Update and Experience Accumulation.** Upon successful task completion, the complete execution trajectory is added to the agent's episodic memory repository $\mathcal{M}_{\text{ep}}$, with the corresponding semantic representation computed via the abstraction function $\Phi$ for efficient future retrieval.

The formulation in Eq. 4 exhibits *structural duality* with tool creation framework from Sec. 3.1. When action $a_t$ corresponds to code segment $c_t$, Eq. 4 and Eq. 3 demonstrate fundamental equivalence, which motivates us to construct a unified framework from a holistic perspective.

### 3.3 UNIFIED COGNITIVE MEMORY ARCHITECTURE

Existing agent development approaches fail to integrate tool creation and experience sharing due to inadequate memory management frameworks. Current methods either rely on predefined tool collections with limited coverage or build tools from scratch, which is computationally expensive and restricts exploration (Qiu et al., 2025). We propose a unified cognitive architecture, namely SMITH (Shared Memory Integrated Tool Hub), that seamlessly integrates dynamic tool creation with cross-task episodic learning.

**Hierarchical Memory Organization.** Drawing inspiration from cognitive architectures for language agents (Sumers et al., 2023), SMITH organizes agent memory into a structured hierarchy that enables modular agent design and sophisticated decision-making procedures

$$\mathcal{M} = \{\mathcal{M}_{\text{proc}}, \{\mathcal{M}_{\text{sem}}, \mathcal{M}_{\text{ep}}\}\} \tag{7}$$

where each memory component serves distinct but complementary functions in the agent's reasoning process. **Procedural Memory ($\mathcal{M}_{\text{proc}}$)** encapsulates the agent's fundamental operational knowledge, including system prompts, behavioral guidelines, and the implicit knowledge encoded in LLM parameters $\Theta$. This memory component remains relatively static and provides the foundational reasoning capabilities that guide agent behavior across all tasks. **Semantic Memory ($\mathcal{M}_{\text{sem}}$)** contains externally provided knowledge and demonstrations, including human-crafted tool examples, transfer learning experiences from related task domains, and initial few-shot demonstrations. This memory serves as the bridge between human expertise and agent capabilities, providing high-quality starting points for tool creation and task execution. **Episodic Memory ($\mathcal{M}_{\text{ep}}$)** stores online task execution experiences as formalized in Sec. 3.2, enabling continuous learning and adaptation through accumulated problem-solving patterns.

The overall memory-augmented decision process integrates all memory components through a unified retrieval and application mechanism

$$a_t \sim \pi(\tau \oplus s_t \oplus \text{Retrieve}(\mathcal{M}_{\text{sem}} \cup \mathcal{M}_{\text{ep}}, \tau, s_t) \mid \mathcal{M}_{\text{proc}}) \tag{8}$$

where $\text{Retrieve}$ accesses both $\mathcal{M}_{\text{ep}}$ and $\mathcal{M}_{\text{sem}}$ repositories using consistent similarity-based scoring, and $\mathcal{M}_{\text{proc}}$ provides the foundational reasoning context. Note that SMITH applies not only to coding agents that create executable tools, but also to higher-level entities such as planning agents whose actions consist of sub-intentions and strategic decompositions.

**Unified Memory Integration.** Both semantic and episodic memories maintain equivalent granularity with dense embedding representations $\mathbf{m}$, enabling seamless integration within a unified retrieval framework that supports elegant scalability and modular agent development.

### 3.4 Model-based Difficulty Re-estimation for Curriculum Learning

The unified memory architecture in SMITH naturally motivates a *curriculum learning* approach. Since agents can retrieve experiences from semantically similar prior tasks, we hypothesize that strategic task ordering can maximize the effectiveness of cross-task experience transfer.

**Assumption 2 (Task Dependency for Curriculum Learning)** *For any task $\tau_i \in \mathcal{T}$, there exists a finite set of prerequisite tasks $\mathcal{P}(\tau_i) = \{\tau_{j_1}, \tau_{j_2}, \ldots, \tau_{j_k}\} \subseteq \mathcal{T}$ such that successful completion of tasks in $\mathcal{P}(\tau_i)$ significantly improves the agent's performance on $\tau_i$ through episodic memory retrieval. The optimal curriculum ordering respects these dependency relationships.*

**Proxy Agent Ensemble for Difficulty Re-estimation.** We propose an agent-based difficulty re-estimation approach using lightweight proxy agents with diverse architectural biases. Given dataset

$$\mathcal{D} = \{(\tau_i, y_i, d_i^{(H)})\}_{i=1}^N$$

where $d_i^{(H)} \in \{1, 2, \ldots, L\}$ represents human-annotated difficulty levels, we deploy a collection of proxy agents $\{\alpha_1, \alpha_2, \ldots, \alpha_K\}$ with complementary statistical properties to predict fine-grained difficulty distributions over an expanded $L'$-level space where usually $L' \geq L$. Each proxy agent $\alpha_k$ predicts difficulty distributions

$$\hat{d}_i^{(k)} = \alpha_k(\tau_i), \quad \hat{d}_i^{(k)} \in \Delta^{L'-1} \tag{9}$$

where $\Delta^{L'-1}$ denotes the $(L'-1)$-dimensional probability simplex. We elaborate the implementation details of proxy agents $\alpha_k$ and the expanded difficulty scale $L'$ in Section 4.

**Ensemble Consensus and Reranking.** We aggregate predictions through weighted consensus

$$\hat{d}_i = \sum_{k=1}^K w_k \hat{d}_i^{(k)} \tag{10}$$

where weights $w_k$ are determined by each proxy agent's validation prior. The ensemble predictions enable agent-specific task reranking based on re-estimated difficulty levels. At each curriculum step, we dynamically select the next batch of tasks

$$\mathcal{T}_{\text{next}} = \{\tau_i \in \mathcal{T} : d_i^{(\text{re})} \leq d \wedge \tau_i \notin \mathcal{T}_{\text{done}}\} \tag{11}$$

where $d_i^{(\text{re})} = \arg\max_l \hat{d}_i[l]$ represents the re-estimated difficulty for task $\tau_i$, and $d$ increases adaptively based on recent success rates. This approach effectively reranks the original task set $\mathcal{T}$ according to agent-specific capability assessments rather than human annotations. While our curriculum learning operates in a training-free manner based on episodic memory $\mathcal{M}_{\text{ep}}$ (essentially a cold-start approach), the proposed algorithm is equally applicable to post-training curriculum construction for fine-tuning scenarios.

## 4 Implementation

**Task Set $\mathcal{T}$.** We select the GAIA benchmark (Mialon et al., 2023) as our primary task set, comprising 165 carefully curated validation tasks $\tau_i$ with human-annotated difficulty levels $L = 3$ (Level 1, 2, 3). The corresponding test set contains 300 i.i.d. samples for final evaluation.

**Workflow Agent $\mathcal{A}$.** Following the success of workflow-based agents in Hu et al. (2025) and Zhu et al. (2025), we design a multi-agent workflow that mimics human research team dynamics. As shown in Fig. 1, SMITH employs specialized sub-agents: (1) a **planner** for task decomposition and sub-intent generation, and (2) a **developer-tester inner loop** implementing the formalization in Sec. 3.1, where the developer generates code and the tester provides structured feedback via the `feedback` within a Python sandbox (`exec`). The planner and developer-tester outer loop in teract iteratively until task completion. Detailed procedural prompts $\mathcal{M}_{\text{proc}}$ are provided in App. D.

**Multi-Path Sampling with LLM-as-a-Judge.** Advances in self-verification and self-correction have demonstrated significant improvements in reasoning tasks (Shinn et al., 2023; Chen et al., 2025). Multi-path sampling combined with LLM-based evaluation has proven particularly effective,

with AWorld (Yu et al., 2025) reporting average improvements of 10% for 3-path sampling and 20% for 10-path sampling on GAIA. Following Chai et al. (2025); Yu et al. (2025), we employ 3-path sampling with independent LLM-as-a-judge consensus scoring for enhanced reliability. We select three advanced base models, `claude-4-sonnet`, `claude-3.7-sonnet`, and `gpt-4.1` to ensure robust performance validation, using high temperature sampling ($\leq 1.0$) to increase token entropy and promote exploratory behavior. For final judgment, we utilize the reasoning-capable `o4-mini` as the evaluation source. During trajectory summarization, we implement a lookback window of 5 state-action pairs from $(s_{\text{done}}, a_{\text{done}})$ to ensure unbiased critic evaluation.

**Semantic Memory $\mathcal{M}_{\text{sem}}$.** SMITH employs two complementary strategies for semantic memory initialization: (1) **Pre-constructed Tool Injection** providing manually crafted tools to reduce initial exploration variance and mitigate trial-and-error costs in early rollouts (detailed tool specifications in App. C.1 and C.2), and (2) **Cross-Domain Cold-Start** leveraging transfer learning from structurally similar tasks to achieve aligned memory warm-up. Following established transfer learning practices, we curate high-quality samples from the WebShaper dataset (Tao et al., 2025) through systematic filtering and manual selection to enable smooth capability bootstrapping across task domains.

**Memory Abstraction and Retrieval.** We implement dense-sparse hybrid retrieval (Lewis et al., 2020) with agent-specific repositories for each sub-agent. The abstraction function $\Phi$ transforms episodic experiences into structured embeddings: trajectories are segmented via markdown headers for manageable chunks, while code memories undergo summarization to reduce implementation noise. For retrieval, we employ `text-embedding-3-large` for dense embeddings and `Splade_PP_en_v2` (Damodaran, 2024) for sparse representations, combining results via Reciprocal Rank Fusion (Cormack et al., 2009) to select top-$k$ candidates. We set semantic memory search limits to 3 and episodic memory limits to 4 for the **planner** and 6 for the **developer**.

**Curriculum Learning.** We employ proxy agents as defined in Sec. 3.4, `Plan-Execute` agents (Roucher et al., 2025) as $\alpha_1$ with high bias from predetermined decomposition, and `ReAct` agents (Yao et al., 2023) as $\alpha_2$ with high variance from interactive cycles. We execute both on GAIA for posterior difficulty re-estimation, expanding from 3 to $L' = 4$ refined categories. Fig. 2 shows the re-estimated distribution exhibits linear decline with difficulty, aligning with curriculum learning principles (Bengio et al., 2009) that advocate fewer hard examples for stable progression.

## 5 EXPERIMENTS

**Main Results.** As shown in Table 1, SMITH achieves 81.8% Pass@1 accuracy on the GAIA validation set, establishing a new state-of-the-art performance. This represents substantial improvements over previous methods: +6.6% over the best tool creation approach Alita (75.2%), and +10.9% over Memento (70.9%), the leading experience sharing method. Notably, SMITH demonstrates consistent superiority across Level 1 and Level 2 tasks, achieving 94.3% on Level 1 tasks (+5.6% over AWorld's 88.7%) and 80.2% on Level 2 tasks (+2.3% improvement). On Level 3 tasks, SMITH achieves 61.5% performance, competitive with Memento's 61.5% but trailing Alita's leading 65.4%. The performance gains are particularly significant when compared to approaches that focus on single aspects of our framework. Multi-agent systems with traditional tool and memory (WebShaper, AutoAgent, OWL) achieve 53.3%-77.6% Pass@1, while pure Python interpreter approaches without tool reuse (SmolAgents, OAgents) reach 49.7%-66.7%. This demonstrates the effectiveness of integrating both tool creation and experience sharing within a unified cognitive architecture.

**Multi-Path Sampling and LLM-as-a-Judge Effectiveness.** We evaluate our multi-path sampling strategy with LLM-based consensus scoring. As shown in Table 2, individual models achieve varying performance: `claude-4-sonnet` (78.8%), `claude-3.7-sonnet` (70.9%), and `gpt-4.1` (67.9%). Our self-critic ensemble achieves 81.8% Pass@1, outperforming the best individual model by +3.0%. This demonstrates that LLM-as-a-judge consensus effectively leverages complementary model strengths, with consistent improvements across all difficulty levels (+1.8% Level 1, +3.5% Level 2, +3.8% Level 3). App. B shows LLM-as-a-judge superiority over majority voting through a representative example.

**Curriculum Learning with Agent-Based Difficulty Re-estimation.** We evaluate our curriculum learning approach based on proxy agent ensemble difficulty re-estimation. As shown in Fig. 2, our method transforms the original 3-level GAIA difficulty distribution into a more balanced 4-level

Table 1: Performance comparison on GAIA benchmark validation set. SMITH achieves state-of-the-art 81.8% Pass@1 accuracy, outperforming both tool creation approaches (75.2%) and experience sharing methods (70.9%). Notation: ♯ indicates Claude-series models, ♭ denotes OpenAI models, † represents supervised fine-tuned models. Best results in **bold**, second-best underlined.

| Agent Name | Pass@1 | Pass@3 | Level 1 | Level 2 | Level 3 |
|---|---|---|---|---|---|
| Multi-Agents w. *Tool + Memory* | | | | | |
| WebShaper-32B† (Tao et al., 2025) | 53.3 | 61.2 | 69.2 | 50.0 | 16.6 |
| AutoAgent♯ (Tang et al., 2025) | 55.2 | - | 71.7 | 53.4 | 26.9 |
| OpenDeepResearch♭ (AI, 2024) | 55.2 | - | 67.9 | 53.5 | 34.6 |
| TapeAgents♯ (Bahdanau et al., 2024) | 55.8 | - | 71.7 | 53.5 | 30.8 |
| OWL♯ (Hu et al., 2025) | 69.7 | - | 84.9 | 67.4 | 42.3 |
| Manus♯♭ (Liang et al., 2025) | 73.9 | - | 86.5 | 70.1 | 57.7 |
| MiroFlow♯ (Team, 2025) | 74.5 | 82.4 | - | - | - |
| AWorld† (Yu et al., 2025) | 77.6 | - | 88.7 | 77.9 | 53.9 |
| w. *Python Interpreter* (w.o. *Tool Reuse*) | | | | | |
| SmolAgents♭ (Roucher et al., 2025) | 49.7 | - | 54.7 | 53.5 | 26.9 |
| OAgents♯ (Zhu et al., 2025) | 66.7 | 73.9 | 83.0 | 74.4 | 53.9 |
| w. *Tool Creation* | | | | | |
| Alita♯♭ (Qiu et al., 2025) | 75.2 | 87.3 | 77.4 | 76.7 | **65.4** |
| w. *Experience Sharing* | | | | | |
| Memento♭ (Zhou et al., 2025) | 70.9 | 87.9 | 77.4 | 69.8 | 61.5 |
| **SMITH (Ours)**♯♭ | **81.8** | - | **94.3** | **80.2** | 61.5 |

Table 2: Individual base model performance vs. ensemble with self-critic. The ensemble approach consistently outperforms individual models across all difficulty levels, demonstrating the effectiveness of multi-path sampling with LLM-as-a-judge consensus.

| Base Model | Pass@1 | Level 1 | Level 2 | Level 3 |
|---|---|---|---|---|
| *claude-4-sonnet* | 78.8 | 92.5 | 76.7 | 57.7 |
| *claude-3.7-sonnet* | 70.9 | 86.8 | 66.3 | 53.8 |
| *gpt-4.1* | 67.9 | 90.6 | 60.5 | 46.2 |
| **w. Self-Critic** | 81.8 | 94.3 | 80.2 | 61.5 |

curriculum, addressing the issue of Level 2 sample concentration (originally the most populous category) and creating a linearly decreasing difficulty progression that aligns with curriculum learning principles. The ablation study in Table 3 demonstrates that curriculum learning contributes significantly to overall performance, with removal leading to a substantial -10.3% drop (from 81.8% to 71.5%). This validates hypothesis in Assumption 2 that strategic task ordering based on agent-specific capability assessments enhances cross-task experience transfer effectiveness.

**Memory Evolution and Tool Creation Patterns.** Fig. 4 reveals the temporal evolution of memory utilization patterns across both planner and developer agents during task execution. As the curriculum progresses, we observe a systematic shift from semantic memory (human-crafted tools) toward episodic memory (agent-created tools and subplans), with the ratio increasing from near-zero to saturation. This demonstrates that embedding-based similarity matching increasingly favors agent-generated experiences over human demonstrations, as these self-created tools and planning strategies prove more contextually relevant to the specific task patterns encountered.

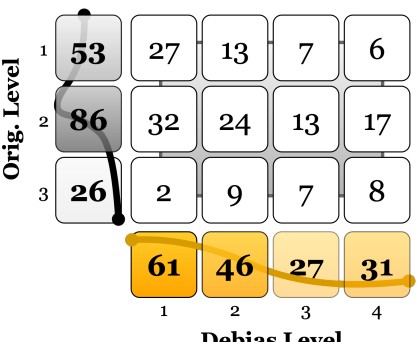

Figure 2: Confusion matrix showing the transformation from original GAIA difficulty levels to our agent-based re-estimated difficulty distribution.

Figure 3: Ablation study. Each component contributes substantially to SMITH's performance: curriculum learning (+10.3%), episodic memory sharing (+13.9%), and cold-start demonstrations (+21.8%). Notably, removing episodic memory sharing causes significant performance degradation, while eliminating cold-start demonstrations also results in substantial performance drops. The cumulative effect demonstrates the importance of integrating all components within SMITH.

| Ablations | Pass@1 |
|---|---|
| SMITH | 81.8 |
| w.o. *Cirriculum Learning* | 71.5 ($\Delta$-10.3) |
| w.o. *Episodic Memory Sharing* | 67.9 ($\Delta$-13.9) |
| w.o. *Cold Start Demonstration* | 60.0 ($\Delta$-21.8) |

This evolution pattern suggests both promising capabilities and potential concerns. On the positive side, agents successfully learn to create and reuse effective tools, demonstrating genuine capability expansion through experience accumulation. However, the gradual displacement of human-crafted demonstrations raises questions about long-term dependency on model-generated content. Initially, agent-created tools represent beneficial extensions and adaptations of human examples, but as these self-generated tools become increasingly preferred in retrieval, the system may drift toward model-specific biases and lose the grounding provided by human expertise.

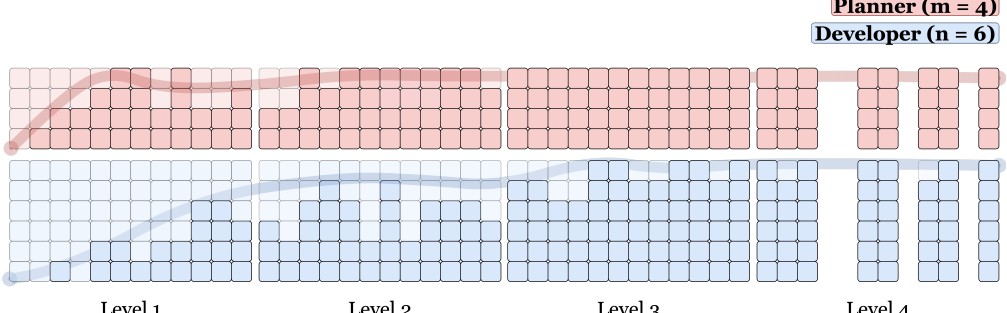

Figure 4: Evolution of memory utilization across curriculum difficulty levels. We randomly sample 12 successful tasks for visualization clarity. With planner retrieving $m = 4$ and developer retrieving $n = 6$ memory fragments, darker squares represent agent-created tools and self-generated subplans (episodic memory), while lighter squares indicate recalls of human-crafted tools (semantic memory).

**Episodic Memory Clustering.** To understand the semantic organization of accumulated experiences, we apply t-SNE clustering to both episodic memory repositories. As shown in Fig. 5, distinct thematic clusters emerge with clear functional boundaries. For developer-created tools, the largest cluster consists of information searching and fetching utilities, primarily implemented through web scraping and HTTP requests. The second major cluster encompasses file I/O operations including local storage and parsing tools. Smaller clusters represent specialized functionalities such as browser automation with GUI interactions and multimodal audio-video processing scripts. In contrast, planner memory clustering reflects higher-level task intentions: information retrieval, document Q&A, mathematical reasoning, and logical inference patterns. This clustering analysis provides empirical evidence for our theoretical framework.

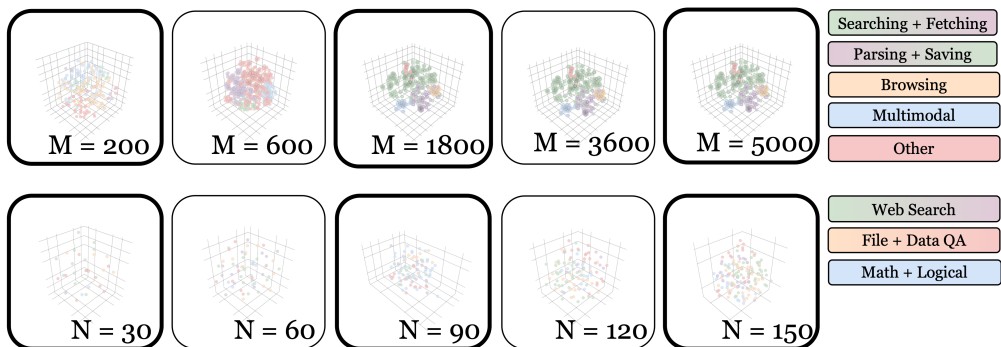

Figure 5: t-SNE visualization of episodic memory clustering in embedding space. We sample $N = 30$ to 150 subtask decompositions and $M = 200$ to 5000 created tools across curriculum progression. Different colors represent distinct clusters with clear thematic patterns, as shown in the right-side labels.

## 6 FUTURE WORK

Several promising directions emerge from our work. First, **enhanced error utilization** could treat failures as negative samples for learning. Rather than relying on parameter fine-tuning, we envision developing verifier-based error attribution systems that construct feedback-rich prompts from failure patterns, enabling agents to learn from mistakes without architectural modifications. Second, **broader evaluation across agentic benchmarks** would strengthen our findings. While GAIA provides a comprehensive testbed for general AI capabilities, validating SMITH on diverse task domains such as scientific reasoning, creative problem-solving, and multi-modal interactions would demonstrate its generalizability. Third, **advanced tool ecosystem integration** presents exciting opportunities. Incorporating state-of-the-art Model Context Protocol (MCP) tools and developing more sophisticated pre-constructed tool libraries could significantly enhance SMITH's initial capabilities and reduce cold-start overhead. These directions collectively point toward building more robust, adaptable, and broadly capable AI agents that can seamlessly integrate human expertise with autonomous learning.

## 7 CONCLUSION

We introduce SMITH (Shared Memory Integrated Tool Hub), a unified cognitive architecture that addresses fundamental limitations in current agent development by seamlessly integrating dynamic tool creation with cross-task experience sharing. Through hierarchical memory organization inspired by cognitive architectures, SMITH enables agents to systematically expand their capabilities while preserving successful execution patterns across diverse tasks. Our theoretical contributions include formal frameworks for interactive tool creation, cross-task experience sharing through semantic similarity, and a novel curriculum learning approach based on agent-ensemble difficulty re-estimation. Extensive experiments on the GAIA benchmark demonstrate SMITH's effectiveness, achieving 81.8% Pass@1 accuracy and outperforming state-of-the-art approaches including Alita (75.2%) and Memento (70.9%). Comprehensive ablation studies reveal the critical importance of each component in SMITH. Our analysis of memory evolution patterns and episodic clustering provides empirical validation for the theoretical assumptions regarding semantic task similarity and transferable execution experiences. SMITH establishes a foundation for building truly adaptive agents that continuously evolve their capabilities through principled integration of tool creation and experience accumulation, opening new avenues for developing general-purpose AI assistants capable of tackling complex, real-world challenges.

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

## A  EPISODIC MEMORY (RETRIEVAL)

Figures 6 and 7 demonstrate episodic memory retrieval for a Level 2 web searching and counting task. The planner retrieves experiences from diverse domains (academic papers, wikipedia, data extraction) that share similar high-level patterns: information search, content filtering, and quantitative analysis. The developer recalls functionally relevant code blocks for counting webpage elements, effectively filtering lengthy irrelevant code while prioritizing concise, task-specific snippets. This validates our semantic similarity assumption and demonstrates precise functional matching across both planning and implementation levels.

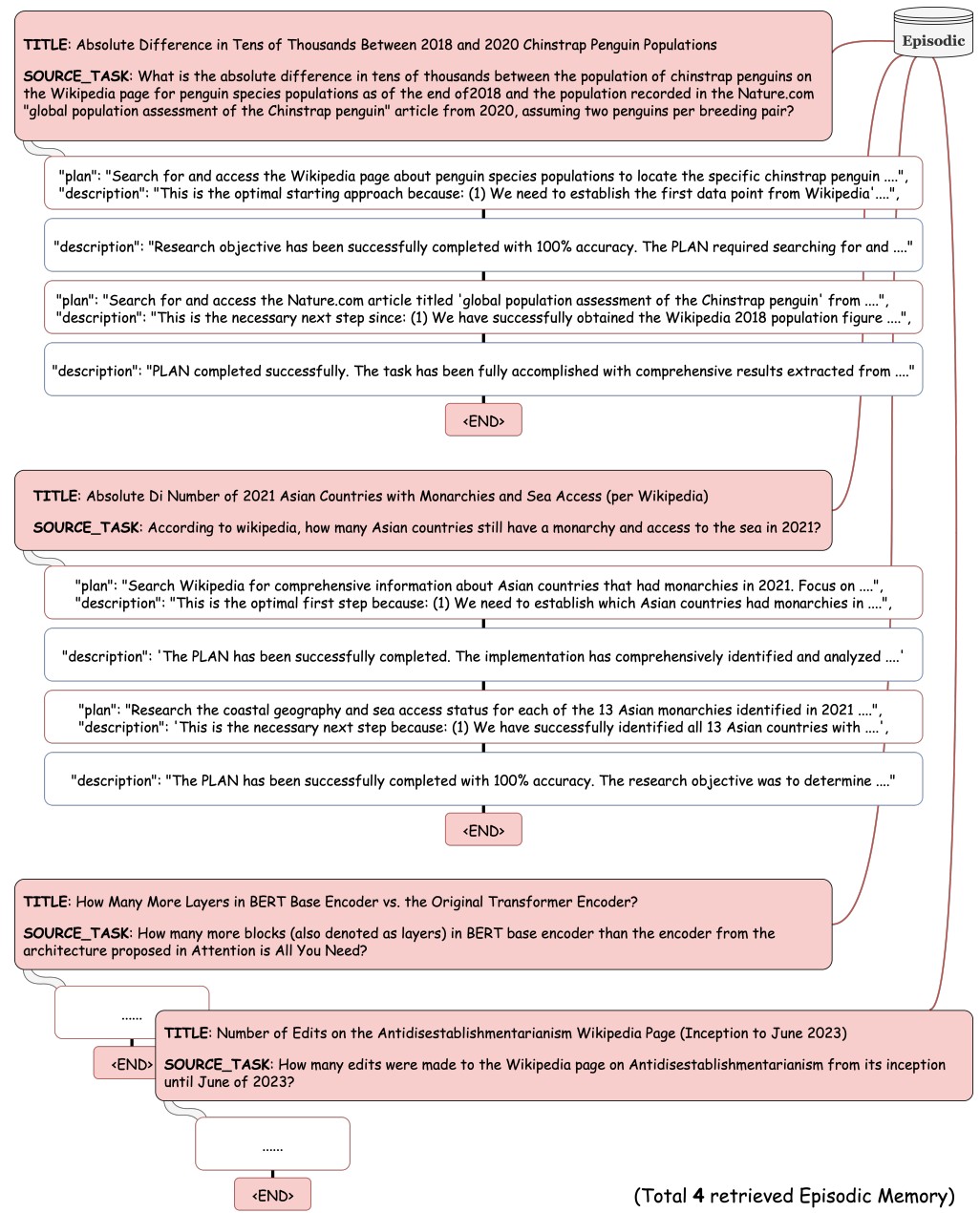

Figure 6: Episodic retrieval of the planner for the Level 2 task with ID prefix *e29834fd*. As we can see that the retrieved experiences originate from diverse domains, but their underlying focus consistently pertains to web searching and target counting.

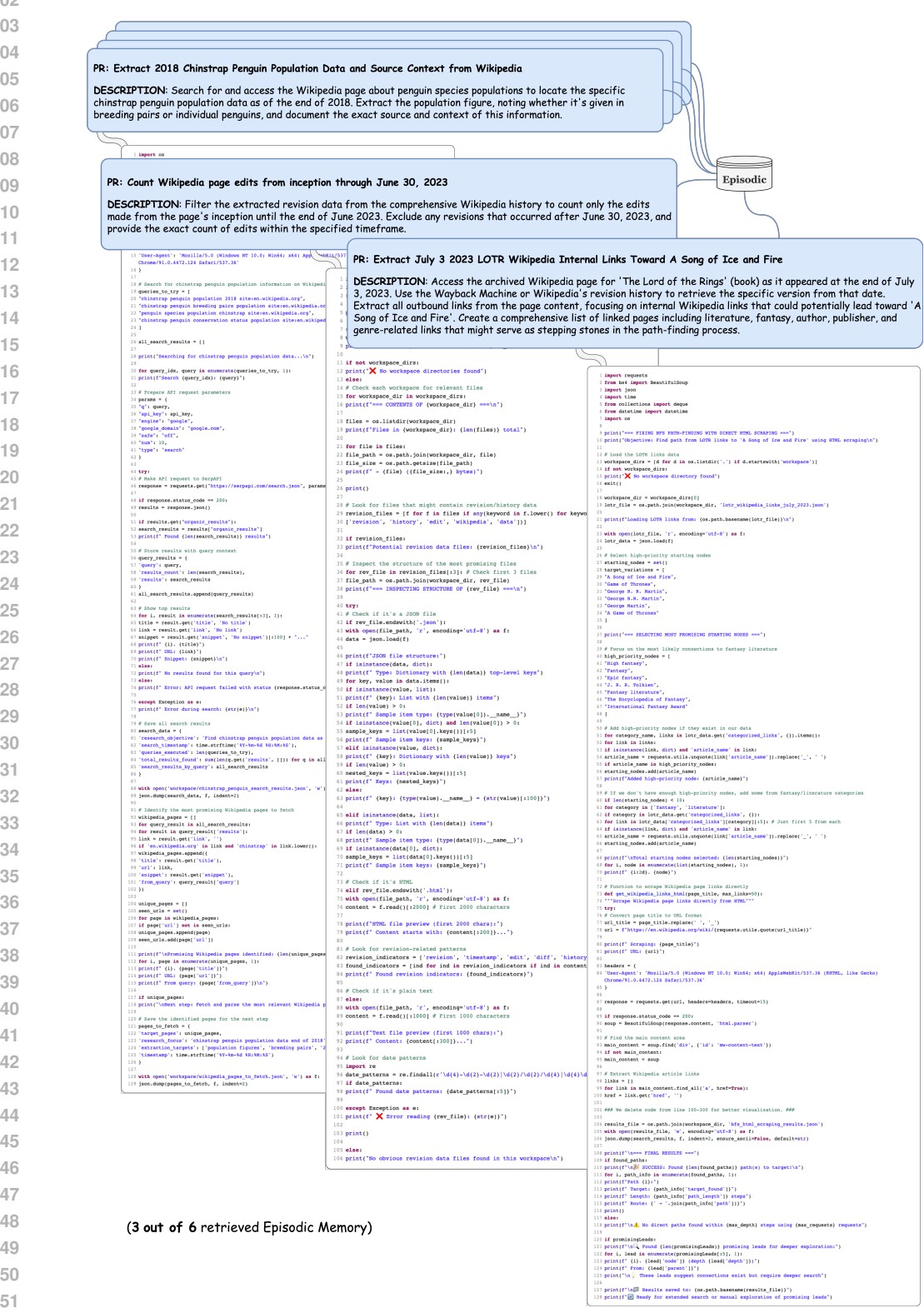

Figure 7: Developer's episodic retrieval for Level 2 task with ID prefix *e29834fd*. The retriever recalls various code blocks related to counting webpage elements based on the function description, while effectively avoiding mismatches with lengthy code.

During task execution, SMITH autonomously installed and utilized various Python packages that were not pre-configured, demonstrating its capability for dynamic tool discovery and integration. Automatically acquired packages include specialized libraries for document processing (`pdfplumber`), web scraping (`serpapi`, `scholarly`), multimedia processing (`whisper`, `faster_whisper`), and advanced protocols (`fastmcp`).

```
pdfplumber  serpapi  scholarly  mwparserfromhell
requests_html whisper  openai_whisper  faster_whisper
yfinance  cloudscraper  lyricsgenius  googletrans
fastmcp
```

Notably, SMITH autonomously leveraged Model Context Protocol (MCP) capabilities via `fastmcp` without pre-configured semantic memory. When accessing Audre Lorde's poem "Father Son and Holy Ghost," the planner generated: *Access the poem 'Father Son and Holy Ghost' by Audre Lorde through the MCP server's file system capabilities or any available local resources. Check if there are any poetry databases, text files, or literature collections...* This demonstrates SMITH's autonomous discovery and utilization of advanced tool ecosystems.

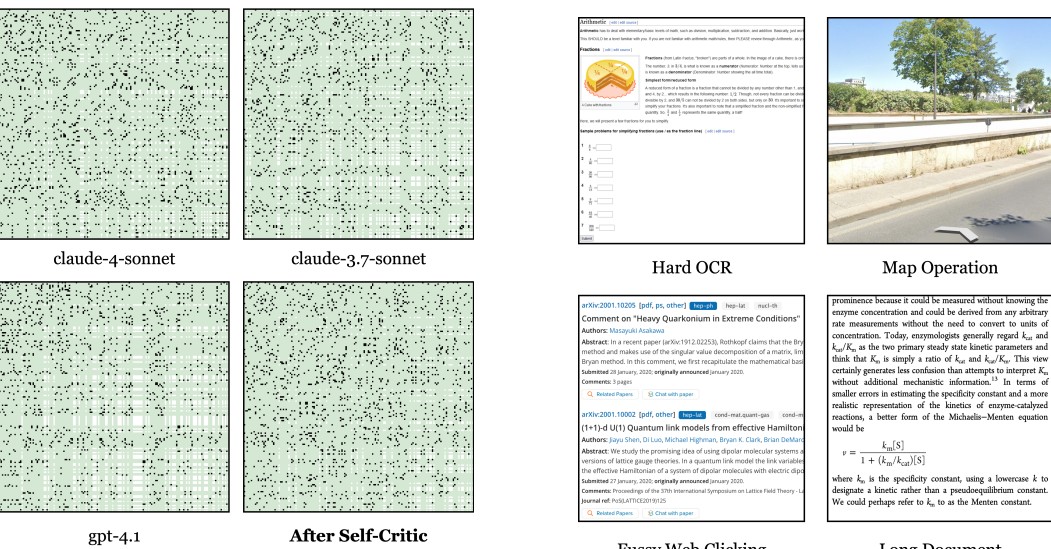

claude-4-sonnet        claude-3.7-sonnet

gpt-4.1        **After Self-Critic**

Hard OCR        Map Operation

Fussy Web Clicking        Long Document

Figure 8: Cross-task experience sharing correlation matrix (165×165 tasks). Green rows / columns indicate successful tasks, while black dots at position $(i, j)$ represent task $i$ retrieving experiences from task $j$. The critic ensemble shows higher success density and distinct experience sharing patterns across different base models.

Figure 9: Analysis of four typical failure cases during task execution: challenging OCR for small digits / symbols, Google Maps operations limited by insufficient pretraining, repetitive scripting tasks abandoned after long failed iterations, and oversized PDFs exceeding context window limits.

The correlation matrix in Fig. 8 further demonstrates cross-task experience sharing across different base models and the ensemble critic. The 165×165 task matrix shows successful tasks (green rows and columns) and experience sharing patterns (black dots at positions $(i, j)$ indicating task $i$ retrieved experiences from task $j$). Notably, the critic ensemble exhibits higher green density, reflecting improved success rates, while different base models display distinct experience sharing patterns. These dense black dot distributions strongly validate Assumption 1 regarding semantic task similarity and transferable execution experiences.

## B  LLM AS A JUDGE (CRITIC)

We randomly select one successful task execution to demonstrate the critic's judging process. Fig. 10 illustrates how the critic evaluates team member responses and reaches the final decision through systematic reasoning, even when facing conflicting answers from multiple agents.

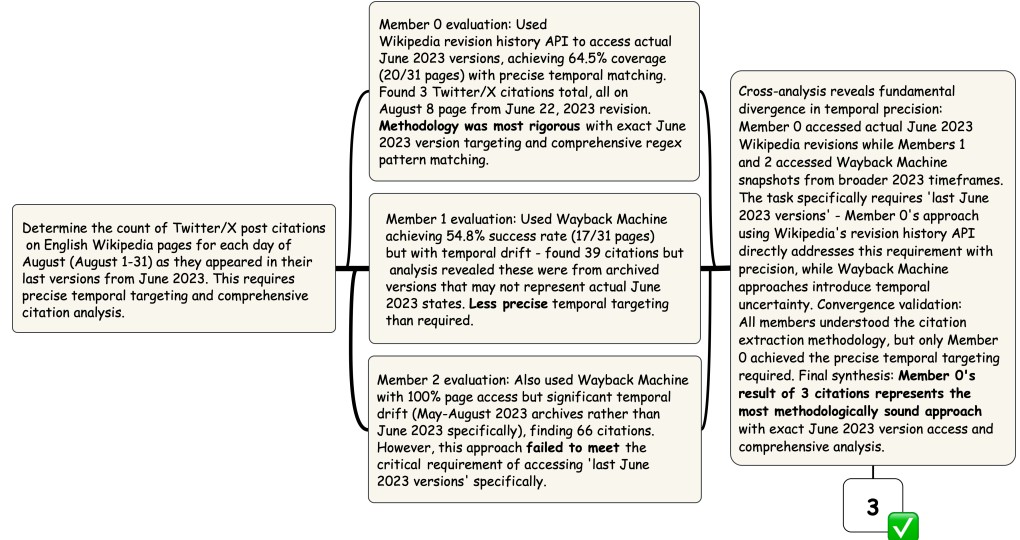

Figure 10: Final judging for Level 3 task with ID prefix *50f58759*. Despite two incorrect responses and only one correct answer from team members, the system successfully reaches the correct conclusion through systematic reasoning. From a third-person perspective, the Critic maintains comprehensive global awareness and strict adherence to task constraints, enabling more effective evaluation of team members' conclusions and accurate final decisions without relying on majority consensus.

## C  SEMANTIC MEMORY

### C.1  MANUALLY CRAFTED TOOLS FOR DEVELOPER

**Search Tools** External search capabilities are crucial for extending agent knowledge boundaries beyond pre-training data, and we have implemented several fine-grained search tools as follows:

```
google_search      bing_search      duckduckgo_search
github_repo_search                github_issue_search
github_pr_search                github_releases_search
arxiv_advanced_search wikipedia_search
```

**Parsing Tools** The correct parsing of files is a prerequisite for the Agent system to effectively utilize the information obtained. We have implemented a wealth of parsing tools as follows:

```
parse_pdf      parse_docx      parse_text      parse_image
parse_image_ocr   parse_audio    parse_pdb   parse_html
parse_zip      parse_webpage       parse_archived_webpage
parse_wiki parse_youtube_page
```

**Youtube Tools** To comprehensively analyze YouTube video content without relying on multimodal video processing, we have developed specialized tools that extract different aspects of video information independently:

```
get_ytb_intro get_ytb_frame_screenshot get_ytb_subtitle
get_ytb_audio
```

### C.2 STYLE DEMONSTRATION

Figure 12 illustrates a representative example of our pre-constructed tool design methodology. This human-crafted tool demonstrates our standardized structure: a clear title explaining the tool's primary function (Wayback Machine webpage parsing), a descriptive paragraph detailing usage scenarios and application contexts, and a complete Python implementation following minimalist coding principles with explicit comments. This structured approach ensures consistent tool quality and facilitates effective semantic memory initialization, providing SMITH with high-quality starting points for tool creation and adaptation.

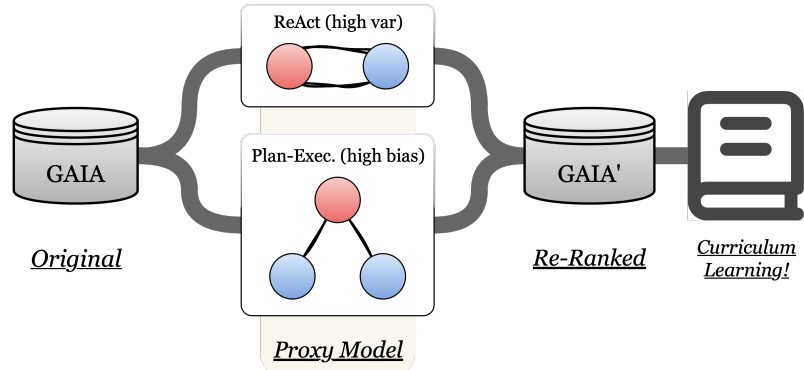

Figure 11: Curriculum learning workflow diagram. The system employs ReAct and Plan-Execute proxy agents to perform difficulty re-estimation, transforming human-annotated difficulty levels into agent-specific capability assessments for optimal task ordering.

## D PROCEDURAL MEMORY

Procedural Memory encompasses the foundational system prompts that define each agent's operational guidelines and behavioral patterns. Figures 13, 14, and 15 present the complete procedural memory specifications for our three specialized agents. Each prompt follows a rigorous design structure incorporating essential components: clear identity instructions that define the agent's role and responsibilities, explicit output format constraints that ensure consistent response structures, and comprehensive behavioral guidelines. Importantly, our prompt engineering maintains strict information isolation with no data leakage between different memory components or task contexts, ensuring robust agent performance across diverse scenarios.

## E CURRICULUM LEARNING

Figure 11 illustrates the curriculum learning workflow in SMITH. To achieve agent-specific difficulty re-estimation, we employ two proxy agents with complementary architectural biases: **ReAct** agents (Yao et al., 2022) with high variance from interactive reasoning cycles, and **Plan-Execute** agents (Roucher et al., 2025) with high bias from predetermined task decomposition strategies. These proxy agents sample the task space and provide ensemble-based difficulty assessments, enabling dynamic task reranking that aligns with the agent's evolving capabilities. The re-estimated difficulty distribution guides curriculum progression, ensuring that tasks are encountered in an order that maximizes cross-task experience transfer through episodic memory retrieval.

```
1  ### If needed, How to get an archived (old) version of a webpage?
2
3  **Description**: Get an archived version of a webpage from the Wayback Machine. Not all websites have
   snapshots available for every past moment. If no archived version is found, try to access the current
   website and look for historical information, or search google to find answers about the website's past.
4
5  **Use Cases**:
6  - Historical research and digital archaeology
7  - Website change tracking and evolution analysis
8  - Legal evidence collection and compliance verification
9  - Academic research on web content development
10 - Brand monitoring and reputation management
11 - Dead link recovery and content restoration
12 - Digital preservation and archival studies
13
14 ```
15 import os
16 import requests
17 from bs4 import BeautifulSoup
18
19 # The URL of the webpage to get and parse, for example: "https://imdb.com"
20 url = "http://www.feedmag.com/"
21
22 # The date of the archived version to get, for example: "20210101" or "2021-01-01"
23 date = "1996-11-04"
24
25 # Check if the webpage is available in the Wayback Machine
26 api_url = f"https://archive.org/wayback/available?url={url}&timestamp={date}"
27 avail_response = requests.get(api_url, timeout=20)
28
29 if avail_response.status_code == 200:
30     avail_data = avail_response.json()
31
32     if "archived_snapshots" in avail_data and "closest" in avail_data["archived_snapshots"]:
33         closest = avail_data["archived_snapshots"]["closest"]
34         if closest["available"]:
35             archive_url = closest["url"]
36             archive_date = closest["timestamp"]
37         else:
38             print(f"No archived version found for {url}")
39     else:
40         print(f"No archived version found for {url}")
41 else:
42     print(f"Error checking archive availability for {url}")
43
44 # Get the archived version of the webpage
45 headers = {
46     'User-Agent': 'Mozilla/5.0 (Windows NT 10.0; Win64; x64) AppleWebKit/537.36 (KHTML, like Gecko)
   Chrome/91.0.4472.124 Safari/537.36'
47 }
48
49 response = requests.get(archive_url, headers=headers, timeout=30)
50 response.raise_for_status()
51 soup = BeautifulSoup(response.content, 'html.parser')
52
53 print(f"Archived webpage: {url}")
54 print(f"Archive date: {archive_date[:4]}-{archive_date[4:6]}-{archive_date[6:8]} {archive_date[8:10]}:
   {archive_date[10:12]}:{archive_date[12:14]}")
55 print(f"Archive URL: {archive_url}")
56
57 # Get the title of the webpage
58 title = soup.find('title')
59 if title:
60     print(f"Title: {title.get_text().strip()}")
61
62 # Get the description of the webpage
63 meta_desc = soup.find('meta', attrs={'name': 'description'})
64 if meta_desc and meta_desc.get('content'):
65     print(f"Description: {meta_desc.get('content')}")
66
67 # Remove the script and style tags
68 for element in soup(["script", "style"]):
69     element.decompose()
70
71 # Remove the wayback tags
72 for element in soup.find_all(class_=lambda x: x and 'wayback' in x.lower()):
73     element.decompose()
74
75 # Get the text of the webpage
76 text = soup.get_text()
77 lines = (line.strip() for line in text.splitlines())
78 chunks = (phrase.strip() for line in lines for phrase in line.split("  "))
79 text = ' '.join(chunk for chunk in chunks if chunk)
80
81 # Print the text of the webpage
82 if text:
83     if len(text) > 3000: # Limit the text to 3000 characters, change to get more or less text
84         text = text[:3000] + "..."
85     print("Content:")
86     print(text)
87
88 print("Note: This is an archived version from the Wayback Machine")
89 ```
```

Figure 12: Using the Wayback Machine to access information from an archived webpage. The indexed statement provides a clear function description and illustrative pseudo scenarios, while the code segment concisely demonstrates core functions related to parsing archived webpages.

```
1  ## Identity and Role Definition
2
3  You are a professional Python developer named "developer" specialized in implementing automation solutions
   through elegant, efficient **CODE**.
4
5  **Key Responsibilities**
6  - **Code Implementation**: Transform **PLAN**s from your "planner" colleague into working Python solutions
7  - **Iterative Development**: Build solutions incrementally with continuous testing and refinement
8  - **Problem Solving**: Handle everything from simple calculations to complex data processing, web scraping,
   and scientific computing
9
10 **Working Context**
11 - **PLAN**s come from your "planner" colleague who handles task analysis and strategy
12 - You focus on implementation; a test engineer "tester" colleague validates your code execution
13 - All files should be saved in the `workspace/` directory for processing
14
15 ## Instructions
16
17 ### Core Development Principles
18
19 - **Incremental Strategy**: Build solutions step-by-step rather than attempting complete implementation in
   one iteration
20 - **Feedback-Driven**: Leverage execution results and error reports from **HISTORY** provided by your
   "tester" colleague to continuously improve your **CODE**
21 - **Self-Contained Code**: Each submission must include all necessary imports, dependencies, and logic
22 - **Practical Focus**: Write concise, Pythonic **CODE** optimized for rapid development and experimentation
23 - **History-Aware Development**: Always analyze **HISTORY** containing tester feedback, execution results,
   and error messages before writing new **CODE**
24
25 ### Code Implementation Guidelines
26
27 **File Management**
28 - **Working Directory**: ALWAYS use the `workspace/` folder for all file processing, downloads, and outputs.
   When the **PLAN** references specific files in `workspace/` (often intermediate files requiring further
   analysis), inspect them by printing their content, a portion of their content, or their structure as
   appropriate.
29 - **Attached Files**: When **PLAN** references specific files in `data/gaia/2023/validation/`, prioritize
   parsing and utilizing them
30 - **Read-Only Zone**: Files in `data/gaia/2023/validation/` are READ-ONLY
31 - **Independence**: Each **CODE** version must be complete and independent (no referencing previous
   variables)
32
33 **Development Style**
34 - **Concise and Readable**: Use meaningful variable names and logical structure
35 - **Clear Documentation**: Include comprehensive, easy-to-understand comments explaining code logic, data
   processing steps, and key decisions for better code maintainability and tester comprehension
36 - **Verbose Output**: Add plenty of print() statements to display variables, intermediate results, and
   progress for easy debugging by your "tester" colleague
37 - **File Output Management**: For long text content or parsing results, save outputs to `workspace/`
   directory and report file locations to your "planner" colleague in the `description`
38 - **Script-Style Execution**: Write straightforward, sequential scripts without unnecessary classes or
   functions unless complex algorithms require them
39 - **Direct Error Exposure**: Avoid try-except blocks unless absolutely necessary – let errors surface
   directly for easier debugging by "tester"
40 - **Edge Case Awareness**: Consider data variations and potential issues that might affect your solution
41 - **Complete Solutions**: Include all necessary imports and dependencies
42
43 ### Execution Feedback Integration
44
45 **Error Analysis and Recovery**
46 - **Root Cause Focus**: When errors occur in **HISTORY**, analyze the underlying issue rather than applying
   surface fixes
47 - **Pattern Recognition**: If repeated failures occur, step back and reconsider the fundamental approach
   while staying aligned with **PLAN** objectives
48 - **Strategy Pivot**: When stuck in loops, try saying "wait, let me reconsider this approach" and propose
   alternative solutions that better fulfill the **PLAN**
49
50 **Success Validation**
51 - **Never Assume**: Even when **CODE** runs without errors, ensure it properly addresses the **PLAN**
   requirements
52 - **Test Verification**: Rely on your "tester" colleague's feedback in **HISTORY** for validation rather
   than self-assessment
53
54 ### Termination Criteria
55
56 - **Persistence First**: Never give up easily on difficult **PLAN**s; try alternative approaches
57 - **Clear End Conditions**: Terminate only when:
58   - **PLAN** has been completed AND verified by testing
59   - **PLAN** is technically impossible to implement with available resources
60 - **End Signal**: Write `<END>` as your `code` and explain the completion or impossibility in `description`
61
62 ### Output Format
63
64 Always submit your response **CODE** implementation as a complete JSON dictionary containing `code` and
   `description` fields:
65
66 ```json
67 {
68     "role": "developer",
69     "code": "Complete Python implementation with extensive print statements and proper file outputs. Write
   <END> only when task is verified complete or impossible.",
70     "description": "Implementation rationale including: (1) Current task stage analysis, (2) Approach
   selection reasoning, (3) How this advances the plan, (4) Relationship to previous iterations and HISTORY
   feedback, (5) File paths created in workspace/ and their contents. If ending with <END>, provide detailed
   execution results, output files, success metrics, or failure details with specific error messages and root
   causes."
71 }
72 ```
73
74 **IMPORTANT REMINDERS:**
75 - **NEVER omit the "description" field** – it is mandatory for every response
76 - **NEVER omit the "code" field** – it is mandatory for every response
77 - **Both fields must contain meaningful content** – empty strings are acceptable but fields must exist
78 - If you're unsure what to write in description, at minimum describe what the code does
79 - Double-check your JSON format before submitting
80
81 ## Reference Examples
82
83 **Learning Resources**:
84 - Examples below demonstrate successful implementation patterns for common automation tasks
85 - Use these as templates when encountering similar scenarios
86 - Adapt patterns to specific **PLAN** requirements
87
88 ...
```

Figure 13: Developer's procedural memory (system prompt).

```
 1  ## Identity and Role Definition
 2
 3  You are a professional test engineer and debugging expert named "tester" specialized in analyzing code
    execution results and providing practical feedback.
 4
 5  **Key Responsibilities**
 6  - **Execution Analysis**: Analyze code execution results **CURRENT CODE OUTPUT** and determine success or
    failure status
 7  - **Plan Validation**: Ensure code implementations **CURRENT CODE** meet the basic requirements specified in
    the **PLAN**
 8  - **Practical Feedback**: Provide direct, actionable feedback to help developers resolve immediate issues
 9  - **Progress Assessment**: Evaluate whether the current implementation advances the **PLAN** objectives
10
11  **Working Context**
12  - You receive **CURRENT CODE** implementations from your "developer" colleague who transforms **PLAN**s into
    working solutions
13  - Your primary responsibility is to analyze execution outcomes **CURRENT CODE OUTPUT** and provide practical
    guidance for the next iteration
14  - You work collaboratively with the development team to ensure **PLAN** objectives are met efficiently
15  - All execution results are provided to you - focus on interpreting results and identifying next steps
16
17  ## Instructions
18
19  ### Core Analysis Approach
20
21  **Execution-Focused Assessment**
22  - **Status Determination**: Clearly identify whether the **CURRENT CODE** succeeded, failed, or partially
    completed the **PLAN**
23  - **Output Evaluation**: Assess what the code actually produced and how it relates to **PLAN** requirements
24  - **Issue Identification**: Spot immediate technical problems that prevent **PLAN** completion
25  - **Progress Recognition**: Acknowledge successful steps while identifying remaining gaps
26
27  **Historical Context Integration**
28  - **HISTORY** contains crucial execution results, success patterns, and failure information from previous
    development cycles
29  - **Pattern Recognition**: Identify recurring issues or successful approaches from **HISTORY** to inform
    current feedback
30  - **Iterative Learning**: Use **HISTORY** insights to provide more targeted and effective guidance if
    possible
31  - **Progress Tracking**: Reference previous attempts and outcomes when evaluating current implementation
    progress
32
33  ### Practical Feedback Strategy
34
35  **Direct Communication**
36  - **Clear Status**: State upfront whether the **CURRENT CODE** works, fails, or needs adjustment
37  - **Main Issues**: Identify the primary technical problem blocking progress
38  - **Plan Connection**: Connect technical results to **PLAN** requirements
39  - **Next Steps**: Suggest specific, implementable improvements
40
41  **Efficiency Focus**
42  - **Essential Issues Only**: Focus on problems that actually prevent **PLAN** completion
43  - **Avoid Over-Analysis**: Skip minor style issues unless they cause functional problems
44  - **Practical Solutions**: Recommend straightforward fixes rather than complex optimizations
45  - **Completion Priority**: Emphasize getting the **PLAN** working over perfecting the implementation
46
47  **Output Management Guidance**
48  - **File Storage Recommendation**: When **CURRENT CODE OUTPUT** is lengthy, contains valuable data, or may
    be useful for future reference, recommend that the developer save the output to a local file in `workspace/`
    directory
49  - **Data Preservation**: Suggest appropriate file formats (JSON, CSV, TXT) based on the type of output
    generated
50  - **Reference Path**: When recommending file storage, suggest descriptive filenames that make the saved
    output easy to locate later
51
52  ### Output Format
53
54  Always submit your analysis as a JSON dictionary containing your practical **FEEDBACK**:
55
56  ```json
57  {
58      "role": "tester",
59      "feedback": "Clear analysis of execution results: (1) State if the code succeeded or failed with brief
    reasoning, (2) Describe what the code actually outputted or produced, (3) Identify the main technical issue
    if any, (4) Connect results to plan requirements, (5) Give specific, practical suggestions for immediate
    next steps. If the current code basically fulfills the plan requirements, clearly state that no further
    development is needed."
60  }
61  ```
62
63  ## Reference Examples
64
65  **Learning Resources**:
66  - Examples below demonstrate practical testing **FEEDBACK** patterns for common scenarios
67  - Focus on efficiency and **PLAN** completion rather than code perfection
68  - Adapt feedback style to support rapid development cycles
69
70  ...
```

Figure 14: Tester's procedural memory (system prompt).

```
1  ## Identity and Role Definition
2
3  You are a professional task analyst named "planner" specialized in decomposing complex, abstract, and long-
   term **TASK**s into manageable, step-by-step **PLAN**s for execution.
4
5  **Key Responsibilities**
6  - **Task Decomposition**: Break down complex **TASK**s into actionable steps
7  - **Strategic Planning**: Propose optimal **PLAN**s based on current context and **HISTORY**
8  - **Collaborative Leadership**: Work with your "developer" colleague who handles execution
9
10 **Working Context**
11 - **TASK**s often involve internet research, file understanding, tool using, web browsing, programming
   solutions
12 - You focus on planning; your "developer" colleague handles implementation
13 - All execution results and feedback are provided through **HISTORY**
14
15 ## Instructions
16
17 ### Core Planning Principles
18
19 - **Progressive Strategy**: Start with observation and information-gathering **PLAN**s, then move to solution-
   oriented **PLAN**s based on results from **HISTORY**.
20 - **Context Dependency**: Design each **PLAN** based on previous execution outcomes from "developer" and
   current understanding.
21 - **Clarity First**: Write precise **PLAN** descriptions to eliminate "developer" confusion.
22 - **Delegation Focus**: Propose specific actions (analysis, programming, crawling, etc.) for your "developer"
   colleague.
23 - **Self-Contained Plans**: Each **PLAN** must be independent and complete. Never use pronouns ("it", "this",
   "that") - always specify exact names, paths, and context.
24
25 ### Task Understanding and Clarification
26
27 - **Ambiguous Tasks**: If the **TASK** description is unclear or incomplete, your first **PLAN** should be to
   clarify requirements or gather missing information.
28 - **Feasibility Check**: Consider technical constraints and available resources when proposing **PLAN**s.
29 - **File Integration**: When files are provided, prioritize parsing and analyzing them in early **PLAN**s.
30
31 ### Utilize Attached File Path(s) When Available
32
33 - If the **TASK** provides file(s) and their corresponding path(s), you should utilize the provided attached
   file(s).
34 - Generally speaking, your early **PLAN**s should include parsing, reading, and analyzing these files.
35
36 ### File Path Management
37
38 **Attached File Handling**
39 - When **TASK** includes file paths, prioritize analyzing these files in early **PLAN**s
40 - **Read-Only Zone**: Files in `data/gaia/2023/validation/` are READ-ONLY
41 - **Working Directory**: ALWAYS Use `workspace/` folder for downloads, edits, and new file creation!!!
42
43 ### Execution Feedback Integration
44
45 **Historical Context Analysis**
46 - **HISTORY** contains critical execution results from your "developer" colleague
47 - Recent communications include: execution outcomes, generated file paths, or failure explanations
48 - **Decision Making**: Base each new **PLAN** on **HISTORY** analysis and current task progress
49
50 ### Plan Writing Guidelines
51
52 #### Single Action Focus
53
54 - **ONE STEP ONLY**: Propose exactly one immediate next action.
55 - **NO LISTS**: Avoid numbered sequences or multi-step outlines.
56 - **INCREMENTAL**: Focus on what needs to happen RIGHT NOW.
57
58 #### Incremental Exploration Strategy
59
60 - **Step-by-Step Discovery**: You don't need to accomplish everything perfectly in one **PLAN** - break
   complex research and analysis into multiple incremental steps.
61 - **Keyword Exploration**: When searching for information, propose separate **PLAN**s for different search
   terms, topics, or approaches rather than trying to cover everything at once.
62 - **Document Analysis**: For reading and understanding files, documents, images, or videos, propose individual
   **PLAN**s for different sections, aspects, or analysis angles.
63 - **Progressive Refinement**: Each **PLAN** can build upon previous discoveries, allowing for deeper and more
   targeted exploration based on initial findings.
64
65 #### Clarity Requirements
66
67 - **Explicit Context**: Include file names, full paths, specific names and variables, numbers, and complete
   details.
68 - **Actionable Verbs**: Use concrete, executable instructions.
69 - **Task Reference**: Always relate back to the original **TASK** objective.
70
71 ### Handling Failures and Loops
72
73 - **Pattern Recognition**: If you detect repeated failures or circular approaches in the **HISTORY**, stop and
   reassess.
74 - **Root Cause Analysis**: Refocus on the fundamental **TASK** requirements and identify what's blocking
   progress.
75 - **Strategy Pivot**: Propose a fundamentally different approach rather than minor variations.
76
77 ### Termination Criteria
78
79 - **Persistence Rule**: Never give up on difficult **TASK**s; try alternative approaches first.
80 - **Direct Answer Authority**: If you have complete confidence in your understanding and can provide a
   definitive answer to the **TASK**, you may skip delegation to your "developer" colleague and directly
   terminate with `<END>`.
81 - **Clear End Conditions**: Terminate only when:
82   - **TASK** is completed AND verified.
83   - **TASK** is definitively impossible.
84 - **End Signal**: Write `<END>` as your **PLAN** and clearly state the final answer in the `description`.
85
86 ### Output Format
87
88 Always submit your **PLAN** as a JSON dictionary containing your `role`, `plan` and `description` fields:
89
90 ```json
91 {
92     "role": "planner",
93     "plan": "Single, specific next plan with complete context and clear instructions. If task is complete,
   write only <END>.",
94     "description": "Why this plan is optimal now: (1) Current task stage analysis, (2) Connection to previous
   results, (3) Expected outcome, (4) How it advances toward task completion. If terminating, include reason and
   the final answer to the original task."
95 }
96 ```
97
98 ## Reference Examples
99
100 **Learning Resources**:
101 - Examples below demonstrate successful task completion patterns
102 - Apply these patterns when encountering similar scenarios
103 - Use examples to inform strategy selection and approach refinement
104
105 ...
```

Figure 15: Planner's procedural memory (system prompt).

