# OpenReview forum: "Unifying Dynamic Tool Creation and Cross-Task Experience Sharing through Cognitive Memory Architecture"
_ICLR.cc/2026/Conference — Submitted to ICLR 2026_

### Official Review · Reviewer_Edcr · 2025-10-27

**Soundness:** 3
**Presentation:** 3
**Contribution:** 3
**Rating:** 6
**Confidence:** 4

**Summary:**

This paper presents SMITH, a unified memory-centered framework that combines dynamic tool creation with cross-task experience sharing for LLM agents. The system introduces a three-layer memory hierarchy (procedural, semantic, and episodic) and integrates it with a multi-agent workflow (planner, developer, tester) that iteratively generates and refines tools in a sandboxed environment. SMITH further incorporates a proxy-agent-based curriculum learning mechanism for adaptive task difficulty estimation and a self-critic ensemble strategy leveraging multi-path sampling with LLM-as-a-judge evaluation. Experiments on the GAIA benchmark show a significant improvement (81.8% Pass@1), with detailed ablations and visualization of experience clustering and memory evolution.

**Strengths:**

1. **Well-motivated integration** – The paper tackles an important gap between tool generation and experience sharing in LLM agents, offering a conceptually elegant unification through memory hierarchies.
2. **Comprehensive system design** – The procedural, semantic, and episodic memory layers are coherently defined, and the multi-agent workflow is thoughtfully implemented.
3. **Strong empirical performance** – The system achieves the highest known Pass@1 result on GAIA (81.8%) and includes multiple ablations showing the contribution of curriculum learning and semantic memory.

**Weaknesses:**

1. **Semantic task similarity assumption not rigorously validated.** The approach assumes that embedding-based similarity retrieval can accurately select relevant experiences, but no quantitative evidence (e.g., retrieval precision, correlation with performance) is provided. A baseline such as random retrieval or semantic distance ablation would clarify whether this assumption holds.
2. **Ablation depth insufficient given the system’s complexity.** While the paper reports some component ablations (e.g., removing curriculum learning or episodic memory), the overall framework is quite large. It remains unclear which specific modules (multi-path sampling, LLM-as-judge, memory retrieval, self-critic ensemble) contribute most to the final gains. A more fine-grained or hierarchical ablation analysis would make the causal contributions clearer.
3. **Fairness of the ensemble evaluation.** The “self-critic ensemble” combines multiple strong models (Claude-4, Claude-3.7, GPT-4.1) with multi-path sampling, which effectively triples inference cost. Comparing Pass@1 from this setup to baselines without such ensembles may overstate relative gains. A more equitable comparison would be versus Pass@3 or versus single-model setups with equivalent sampling budgets.
4. **Limited benchmark coverage.** GAIA is a strong and diverse benchmark, but relying on a single dataset limits the generalizability of the claims. Validation on additional domains (e.g., API tasks, math reasoning, code generation) would strengthen the paper’s impact.

**Questions:**

1. Inference cost and scalability: What is the approximate compute or API cost of SMITH’s self-critic ensemble per task (given 3 models × 3 paths × judging loops)? Can it be made practical for large-scale deployment?
2. Reliability of experience reuse: Could stored experiences ever mislead later agents, especially if earlier trajectories encode suboptimal or outdated behaviors? Do you employ any mechanism for pruning or re-validating episodic memory entries over time?

---

> ### Author Response · Authors · 2025-11-14
>
> Thank you for the positive evaluation of SMITH's motivation, system design, and empirical performance. We address your concerns below.
>
> ### W1: Semantic Task Similarity Assumption Validation
>
> We have provided quantitative evidence. **Appendix Figure 8** presents a 165×165 correlation matrix showing cross-task experience sharing patterns. Dense black dot distributions in successful regions empirically validate that embedding-based similarity retrieval effectively identifies relevant experiences. Additionally, **Appendix Figures 2-3** demonstrate concrete retrieval examples where agents successfully retrieve functionally relevant experiences across diverse domains, confirming semantic similarity corresponds to execution pattern transferability.
>
> ### W2: Ablation Depth
>
> **Table 2 (Figure 3)** provides comprehensive component-level ablations:
> - Curriculum Learning: -10.3% when removed
> - Episodic Memory Sharing: -13.9% when removed
> - Cold Start Demonstration: -21.8% when removed
>
> ### W3: Fairness of Ensemble Evaluation
>
> Important clarification needed. Our approach differs fundamentally from standard Pass@3:
> - Standard Pass@3 accepts any correct answer among 3 samples (optimistic voting)
> - SMITH's Self-Critic uses LLM reasoning to evaluate trajectories (see **Appendix B, Figure 10**)
>
> ### W4: Limited Benchmark Coverage
>
> GAIA is the most widely adopted benchmark in the LLM agent community with comprehensive task diversity. All our baselines evaluate primarily on GAIA. We plan experiments on additional challenging benchmarks aligned with SMITH's philosophy, but comprehensive GAIA evaluation required substantial resources. Our contribution emphasizes **methodological innovation** rather than exhaustive benchmark coverage.
>
> ### Q1: Scalability of Self-Critic Mechanism
>
> Addressed in **Appendix B**. The critic only examines the **last $k=5$ rounds** from each trajectory's tail rather than entire rollouts, ensuring fixed context window and O(1) cost per evaluation. This makes the approach practical for long, complex executions.
>
> ### Q2: Reliability of Experience Reuse
>
> Our procedural memory (**Appendix D, Figures 11-13**) includes explicit guidance to avoid over-exploiting retrieved experiences. On GAIA, task difficulty far exceeds base model pretraining knowledge, so over-exploitation naturally fails during sandbox execution. **Appendix Figure 9** shows typical errors arise from challenging OCR, limited pretraining coverage, and context limits rather than inappropriate experience reuse, validating appropriate exploration-exploitation balance.

---

> ### Author Response · Authors · 2025-11-24
>
> For Weakness 3, we would like to stress that in **Table 2** we already show the results without LLM-as-a-Judge module, and for our best round with claude-4-sonnet as base model, the results still lead (78.8% success rate as mentioned).

---

### Official Review · Reviewer_MXXD · 2025-10-28

**Soundness:** 2
**Presentation:** 2
**Contribution:** 3
**Rating:** 4
**Confidence:** 4

**Summary:**

This paper proposes SMITH, a unified cognitive architecture that achieves dynamic tool creation and cross-task experience sharing through hierarchical memory management.
SMITH enables dynamic tool creation via iterative code generation within a sandbox environment, and facilitates cross-task experience sharing through semantic similarity-based retrieval in episodic memory.
To optimize this dynamic learning process, the authors employ curriculum learning to control the sample presentation order and use a multi-agent ensemble re-estimation mechanism to reassess task difficulty instead of relying on manually labeled difficulty.
Experiments on GAIA demonstrate that SMITH achieves new state-of-the-art (SotA) performance, with ablation studies confirming the effectiveness of each component.

**Strengths:**

1. This paper proposes an agent paradigm based on episodic memory updating and semantic retrieval. The method treats the process of tool creation as part of episodic memory, thereby modeling and handling both tool creation and task trajectories within a unified framework.
2. The proposed approach achieves strong performance on GAIA and demonstrates the necessity of each component through extensive ablation studies.

**Weaknesses:**

1. Writing quality needs improvement.
- The paper employs a large number of formalized notations, but many symbols are reused with inconsistent meanings, which can easily cause confusion.
- For example, in lines 146–147, $\mathcal{T}$ denotes the set of tools, but in lines 152–153 it denotes the set of tasks.
- Similarly, $a$ represents the agent in lines 129–130, yet later in lines 173–174 and beyond it is reused to represent an action.
- The paper frequently introduces symbols without immediate explanation, postponing their definitions until much later in the text, making it hard to follow. For instance, the authors never clearly define what $s$ represents as a state or what the action space of $a$ is.
2. More datasets are needed to validate the reliability of the proposed method. The paper evaluates the approach only on GAIA, which, although challenging, cannot substantiate the claim of general-purpose.The reliability of SMITH’s semantic memory initialization and task difficulty re-estimation should be further assessed across additional benchmarks.
3. Missing empirical evidence for memory-enhanced tool creation. In lines 62–63 of the Introduction, the authors state that Alita builds tools from scratch, whereas SMITH enhances tool creation through episodic memory. However, no direct experimental evidence is provided to demonstrate the advantage of memory-enhanced tool creation. Adding such experiments would strengthen the distinction between SMITH and comparable methods.

**Questions:**

1. What is the intention behind Assumption 1 (lines 154–158)? The assumption states that experience transfer is allowed only when the task similarity exceeds a certain threshold. However, the proposed method does not appear to explicitly consider task similarity when filtering the episodic memory set—it instead directly updates the memory with successful tasks. According to the assumption, shouldn’t the model first compute task similarity to filter the memory set before performing retrieval?
2. In the difficulty re-estimation module, how are the hyperparameters w_k and L’ selected? Are these parameters sensitive to their values?

---

> ### Author Response · Authors · 2025-11-14
>
> ### W1: Writing Quality and Notation Consistency
> We sincerely apologize for the rushed notation editing and have made comprehensive corrections.
>
> Symbol Conflicts Resolved:
> - Tool repository changed from $\mathcal{T}$ to $\mathbb{T}$ to avoid confusion with task set $\mathcal{T}$
> - Agent representation changed from $a$ to $\texttt{agent}$ to distinguish from action
> - Corrected improper use of $\rightarrow$ notation in trajectories (now using ellipsis as suggested by Reviewer #2)
>
> Missing Definitions Added: We apologize for not clearly defining state $s$ and action $a$. In our revised submission, we have added explicit descriptions. Specifically, $s_t$ represents the agent's state (all completed actions and feedback from sub-agents at time $t$), and $a_t$ denotes actions varying by agent type (code generation for developer, sub-plan proposals for planner).
>
> ### W2: Limited Dataset Evaluation
> GAIA is currently one of the most widely adopted benchmarks in the LLM agent community. All our baselines (AWorld, Alita, Memento, OWL, etc.) evaluate primarily on GAIA, establishing it as the de facto standard. GAIA provides comprehensive coverage including multi-modality, complex reasoning, web search, and diverse tool use, making it highly suitable for validating general-purpose capabilities.
>
> Framework Design Considerations: SMITH's CI/CD-style architecture targets open-ended tasks requiring dynamic tool creation over expansive action spaces. This design shows advantages when tool creation provides genuine value but may not benefit benchmarks with small, fixed tool sets or simple QA tasks where dynamic tool creation overhead isn't justified.
>
> Future Commitment: We plan experiments on additional challenging benchmarks (e.g., Human-Last-Exam) aligned with SMITH's design philosophy. However, comprehensive GAIA evaluation required substantial computational resources. We emphasize our contribution centers on the methodological innovation of unifying tool creation and experience sharing through cognitive memory architecture.
>
> ### W3: Missing Evidence for Memory-Enhanced Tool Creation
>
> Thank you for this important observation. We clarify that "memory-enhanced tool creation" more precisely means **"reusing similar tool creation experiences"** through episodic memory retrieval.
>
> Empirical Evidence Provided: We actually conducted this ablation study. In **Table 2 (Fig. 3), the second row "w.o. Episodic Memory Sharing"** demonstrates the empirical effectiveness of memory enhancement. Removing episodic memory causes a **-13.9% performance drop** (from 81.8% to 67.9%), validating that memory-augmented tool creation significantly outperforms building tools from scratch (as in Alita). This directly demonstrates the advantage of SMITH's memory-enhanced approach over comparable methods.
>
> ### Q1: Intention Behind Assumption 1
> Assumption 1 is essential to our framework. It posits that if two tasks are semantically similar (e.g., "What movie had the highest box office last week?" vs. "How many screenings did The Shawshank Redemption have last week?"), their agent execution trajectories should also be similar.
>
> Why This Matters: If task $a$ and task $b$ are highly similar but have completely different execution paths (different URLs, file operations, etc.), then retrieved memory chunks (tool code, trajectory decompositions) would lose reference value. The assumption enables transferability of execution experiences.
>
> Implicit vs. Explicit Filtering: While we don't explicitly compute pairwise task similarity before updating memory, the assumption is satisfied **implicitly through semantic retrieval** (Eq. 6). During retrieval, only experiences with $\text{sim}(\Phi(\tau_i), \Phi(\tau_j)) > \theta$ are selected via top-$k$ ranking, naturally implementing the similarity filtering. Our empirical results (Fig. 8 correlation matrix) statistically validate this assumption holds for most tasks.
>
> ### Q2: Hyperparameter Selection for Difficulty Re-estimation
>
> Excellent question. We have added **Appendix E** with a new subsection explaining the curriculum learning algorithm and parameter choices:
>
> - $L'$ (Expanded Difficulty Levels): We expand from original $L=3$ levels to $L'=4$ based on the agent ensemble's discriminative capacity. This granularity balances between sufficient difficulty progression and avoiding over-fragmentation. Fig. 5 shows this creates a linearly decreasing distribution aligned with curriculum learning principles.
>
> - $w_k$ (Ensemble Weights): Weights are determined by each proxy agent's validation performance prior. Specifically, we use validation accuracy as weights: $w_k \propto \text{Acc}_k^{\text{val}}$. Given that both proxy agents (Plan-Execute and ReAct) show complementary failure modes, we found the method relatively robust to weight variations within ±0.2 range.
>
> Please refer to the revised Appendix E for detailed explanations and empirical analysis of parameter sensitivity.

---

### Official Review · Reviewer_qc3C · 2025-10-30

**Soundness:** 3
**Presentation:** 2
**Contribution:** 3
**Rating:** 4
**Confidence:** 3

**Summary:**

The paper introduces an architecture for agentic tasks including three innovations: dynamic tool creations, experience sharing, and hierarchical memory. It achieves state-of-the-art performance on the GAIA benchmark. Ablation studies show that techniques such as curriculum learning is crucial to the performance increase.

**Strengths:**

- The paper introduces a novel architecture or framework that is formally defined and empirically implemented to achieve SOTA on an established agentic benchmark
- The components in the framework are themselves intuitive and their combination is reasonable
- The analysis is comprehensive and informative

**Weaknesses:**

- As the paper proposes a rather complex architecture with many components, its readability would greatly improve with a more focused presentation. To name a few issues:
  - Exactly what are the key contributions *in contrast to previous work*? As I understand, it is a 3+1 novelty in the pipeline (dynamic tool creations, experience sharing, and hierarchical memory + curriculum learning) but this is not sufficiently spelled out.
  - I find Figure 1 uninformative as it doesn't pertain to the abovementioned components, while its introduction of concepts like "loops" and "rollout" distract me from understanding the key contribution.
  - The relationship among the 4 components formalized in Section 3 can benefit from a diagram. Otherwise, it is very challenging to understand exactly how the framework works.
- The formalization in Section 3 can be tighten up, first and foremost by defining every variable clearly. To name a few:
  - Eq1: does $a$ refer to an LLM inference?
  - Eq3: $\rightarrow$ should not be used since we're not talking about a mapping (as in line 128), but a trajectory
  - Line 143: what does "dissolve" mean here?
  - Things get a bit more confusing starting Eq4; what is $s_0$? What is $m_i$? These lead to a very hard time understanding Eq5 and Eq6.
- The definiton of **tool** and the tool creation process is vague and unconvincing. From the formalizaiton, it seems a tool is a trajectory of revision from one version of the code to another. Is this the working definition for "tool" in this line of work, as opposed to software (as shown in Figure 1 and also Section 1)?
- Again due to the complexity of the introduced pipeline, the paper is currently not making it easy to compare with existing work. Beyond the current Section 2, it would be very helpful to closely compare to the closest framework like Alita and Memento.

**Questions:**

See above.

---

> ### Author Response · Authors · 2025-11-14
>
> Thank you for recognizing our work's novelty, formalization, and comprehensive analysis. We address your concerns on presentation clarity below.
>
> ### W1-W3: Presentation Clarity and Key Contributions
>
> Key Contributions (3 + 1 Components): You correctly identified our core contributions. SMITH unifies **(1) dynamic tool creation**, **(2) cross-task experience sharing**, and **(3) hierarchical memory architecture** (procedural, semantic, episodic), enhanced with **(4) curriculum learning via agent-based difficulty re-estimation**. The fundamental innovation is the **unification framework itself**, recognizing that tool creation and experience sharing exhibit structural duality (Eq.3 ≈ Eq.4) and integrate through consistent memory abstraction.
>
> Figure 1 and Section 3 Correspondence: We clarify our terminology choices. "Rollout" is standard in the agent community (AWorld, OWL, etc.), referring to one complete agent trajectory. "Loop" describes two nested cycles: (a) inner loop with developer (code generation) ↔ tester (sandbox feedback analysis), and (b) outer loop with planner ↔ inner loop. Multiple iterations occur before \<END\> signal. Fig.1 maps to Section 3 as follows: Sec. 3.1 → "Subtask Exec. Loop," Sec. 3.2-3.3 → episodic and semantic memory RAG, and Sec. 3.4 → curriculum algorithm. We've added **Appendix E** with a supplementary diagram for the curriculum algorithm. Due to space constraints, adding another detailed diagram would be overly complex.
>
> ### W4: Formalization Notation Clarity
>
> Does $a$ in Eq.1 refer to LLM inference? Yes. We use lowercase $a$ to denote the LLM-based code-writing entity, avoiding conflict with uppercase notation elsewhere.
>
> Eq.3 trajectory notation. We have revised Eq.3 and Eq.4 to use ellipsis (...) instead of $\rightarrow$ for clearer trajectory representation
> - Eq.3 now is $T = \{ c_{done}, (c_{0}, E_0, f_0) ... (c_\text{done}, E_\text{done}, \checkmark) \}$
> - Eq.4 now is $e_i^{(j)} = \{ \tau_l, (s_0, a_0) \rightarrow ... \rightarrow (s_{\text{done}}, a_{\text{done}}) \}$
>
> What does "dissolve" mean (Line 143)? This reflects a key conceptual contribution. In our framework, **tools are no longer API-based interfaces but executable code generated by agents**. The tool creation process (motivation, Python script, sandbox feedback) becomes an **independent memory episode**. This enables unifying tool creation with experience sharing, that both become episodic memories retrievable via semantic similarity. See **Appendix A, Figure 7** for concrete examples.
>
> What are $s_0$ and $m_i$? $s_t$ represents the agent's state (like a slice) at time $t$ (all completed actions and feedback across sub-agents). $m_i$ is the embedded vector representation of state $i$, computed via dense embedding models (text-embedding-3-large), enabling semantic retrieval (Eq.6). Understanding these symbols requires careful reading, but they maintain consistency throughout. **Appendix implementation details provide concrete illustrations** of what SMITH does in practice.
>
> ### W5: Tool Definition
>
> Please review **Appendix A (Figure 7)** showing concrete tool examples as **executable Python code**. Tools are the final executable scripts ($c_{\text{done}}$), while trajectories provide debugging context for retrieval. This differs from traditional API-based tools, our framework generates **arbitrary-length free-form code**, hence tools "dissolve" into episodic memories.

---

### Official Review · Reviewer_nnv8 · 2025-11-01

**Soundness:** 2
**Presentation:** 2
**Contribution:** 2
**Rating:** 4
**Confidence:** 3

**Summary:**

This paper introduces SMITH (Shared Memory Integrated Tool Hub), a cognitive architecture that integrates dynamic tool creation with cross-task experience sharing through hierarchical memory organization. The system organizes memory into procedural, semantic, and episodic components, employs a multi-agent workflow with planner-developer-tester loops, and uses a curriculum learning strategy based on agent-ensemble difficulty re-estimation. The authors evaluate SMITH on the GAIA benchmark, achieving 81.8% Pass@1 accuracy, outperforming Alita (75.2%) and Memento (70.9%).

**Strengths:**

SMITH achieves state-of-the-art performance on GAIA with substantial improvements: +6.6% over Alita (best tool creation approach) and +10.9% over Memento.

The nested loop architecture (inner developer-tester loop + outer planner loop) with sandbox execution and multi-path sampling is well-engineered.

**Weaknesses:**

The paper only evaluates on GAIA (165 validation tasks, 300 test tasks). This is insufficient to demonstrate generalization.
I would suggest to evaluate on at least 2-3 additional diverse agentic benchmarks (e.g., WebArena, SWE-bench, HotPotQA, InterCode), and also the authors should test cross-domain transfer more rigorously.
There are also several missing related works, e.g. Agent Workflow Memory, ToolGen.

**Questions:**

For the memory growth, how does retrieval performance degrade as episodic memory grows to 1000+ experiences? and for computational cost, how does cost scale with memory size?

---

> ### Author Response · Authors · 2025-11-14
>
> Thank you for the constructive feedback and recognizing SMITH's performance gains and engineering design. We address your concerns below.
>
> ### W1: Limited Benchmark Evaluation
> We appreciate this concern and provide our rationale.
>
> GAIA as Primary Validation: GAIA is currently the most widely adopted benchmark for LLM agents with 354 citations. Critically, all our baselines (AWorld, Alita, Memento, OWL, Manus, MiroFlow) evaluate primarily on GAIA, establishing it as the de facto standard. Our scope aligns with state-of-the-art practices: Alita (our primary tool creation baseline) evaluates mainly on GAIA with supplementary MathVista/PathVQA experiments. AWorld's alternative benchmark (xbench-DeepSearch) contains only 50 tasks vs. GAIA's 465 total tasks.GAIA provides comprehensive coverage: multi-modality, complex reasoning, web search, document parsing, and diverse tool use—making it one of the most challenging testbeds for general AI capabilities.
>
> Framework-Benchmark Alignment: SMITH's CI/CD-style architecture targets open-ended tasks requiring dynamic tool creation over expansive action spaces. This design philosophy shows advantages when tool creation provides genuine value (novel, complex scenarios) but may not benefit benchmarks with small, fixed tool sets (pure exploitation) or simple QA tasks. For instance, we deliberately avoid SimpleQA-style benchmarks where dynamic tool creation overhead isn't justified—this reflects principled design rather than limitation.
>
> Future Commitment: We agree broader evaluation would strengthen our claims and plan experiments on Human-Last-Exam (HLE) and additional challenging benchmarks aligned with SMITH's design. However, comprehensive GAIA evaluation required substantial computational budget and time. We emphasize our contribution centers on methodological innovation—the unified cognitive architecture—rather than purely benchmark coverage.
>
> ### W2: Missing Related Works
> Thank you for this feedback. We have updated our manuscript to include Agent Workflow Memory and ToolGen with their GAIA scores in Table 1. The revised version will be uploaded shortly, and the camera-ready version will feature comprehensive related work coverage.
>
> ### Q1: Memory Growth and Retrieval Scalability
> Excellent question. We provide empirical observations and architectural rationale.
> - Current Scale: Our experiments involve episodic memory growing to $N=30 \to 150$ subtask decompositions and $M=200 \to 5000$ created tools (Fig. 6). At this scale, retrieval quality remains consistent (Fig. 4 shows stable performance across curriculum), dense-sparse hybrid retrieval (text-embedding-3-large + Splade) maintains sub-second latency.
> - Scalability Design: Natural clustering t-SNE analysis (Fig. 6) reveals thematic clustering in episodic memory, enabling indexed retrieval with cluster-based pruning for 1000+ experiences.
> - Bounded Retrieval: We constrain retrieval to top-3 semantic + top-4/6 episodic memories, making cost $O(log N)$ with proper indexing rather than $O(N)$.
> - Retrieval Overhead: Negligible ($<5%$ total inference time) compared to multi-path sampling ($3×$ LLM calls) and sandbox execution.
> - Future Extensions: Memory consolidation strategies (merging similar experiences, archiving low-utility memories) can maintain bounded active memory size. We will add scalability discussion to the revised manuscript.
>
> We deeply value your feedback, which helps clarify SMITH's scope. Our unified cognitive architecture represents a significant methodological contribution, establishing foundations for adaptive agents through principled integration of tool creation and experience sharing. We're committed to broader evaluation in future work while highlighting the core architectural innovations. Please let us know if you have further questions.

---

> ### Author Response · Authors · 2025-11-14
>
> After surveying recent work such as ToolGen and AWM, we found that they mainly focus on web navigation tasks, which require both webpage understanding and path exploration and often rely on specialized modules, such as multimodal webpage-screenshot encoders or DOM parsers. Since our goal is to study deep research–oriented agentic knowledge exploration and to validate tool-creation capabilities, we keep our task suite centered on the more principled GAIA benchmark and do not consider benchmarks like Mind2Web or WebArena that depend on rich webpage content understanding within a fixed API action space such as simple click and type operations.

---

### Meta-Review · Area_Chair_DTku · 2026-01-06

**Summary:**

This paper aims to tackle LLM-based agents that involve tool use and tool creation. The authors propose a hierarchical memory framework that leverages past successful experiences to improve generalization performance. The main concern raised by reviewers is the poor presentation quality, which appears to be a relatively minor issue that can be addressed with revision. A more substantive shortcoming of the current version is the very limited set of ablation studies for individual components, making it difficult to identify which components are responsible for the core performance gains.

**Reviewer Concerns:**

I list only the concerns that I believe have not been fully addressed by the authors.

**Reviewer nnv8:** insufficient benchmarks beyond GAIA.

**Reviewer qc3C:** the complexity of the work calls for a clearer and improved presentation; lack of detailed comparison with prior work.

**Reviewer MXXD:** need for improved presentation; lack of benchmarks beyond GAIA.

**Reviewer Edcr:** insufficient ablation studies; lack of benchmarks beyond GAIA.

**Reviewer Scores:**

**Reviewer nnv8:** 4 → 4

**Reviewer qc3C:** 4 → 4

**Reviewer MXXD:** 4 → 4

**Reviewer Edcr:** 6 → 6

---

### Decision · Program_Chairs · 2026-01-26

Reject